# HYBRIDSKETCHNET: SKETCH-BASED 3D HUMAN MESH RECONSTRUCTION VIA HYBRID POINT-IMAGE NETWORKS

## ABSTRACT

Sketches are an efficient and effective tool for generating 3D human meshes with arbitrary body shapes and poses. However, current mesh reconstruction methods are mainly designed for natural images, which are hard to apply to sketches due to the abstract and sparse characteristics of the latter. Moreover, there is no dataset with sufficient sketch-meshes pairs for developing and evaluating relevant methods. To tackle these issues, we introduce a hybrid framework that fits parametric human models (e.g., skinned multi-person linear model) to sketches in a coarse-to-fine manner. Specifically, the proposed framework consists of three core components: (i) Given a sketch image as the input, a vision transformer-based Local Image Encoder (LIE) is introduced to model the local structures of the sketch and yields a coarse mesh estimation. (ii) A Global Point Encoder (GPE) taking the 2D coordinates of sketch contours as inputs, is also utilized to obtain the global representation of the sketch. (iii) As the local presentation can depict human poses more precisely while the global representation is more suitable for body shapes, we propose a graph-based refiner (GRefiner) to leverage the advantages of both representations and generate the final well-fitted mesh. Furthermore, we collect a large-scale dubbed Sketch3DS, containing approximately 10,000 paired sketches and human meshes with diverse poses and shapes. Extensive experiments on Sketch3DS demonstrate that the proposed approach outperforms existing methods, achieving accurate alignment between input sketches and constructed human meshes.

## 1 INTRODUCTION

Sketching is an effective medium for communicating ideas in art, design, architecture, and engineering Zeleznik et al. (2006). The conversion of sketches into 3D models can significantly improve design efficiency and reduce resource consumption De Paoli & Singh (2015). In particular, sketch-based 3D human modeling has emerged as a promising yet insufficiently explored area.

Unlike natural images, hand-drawn sketches are highly abstract and simplified, lacking detailed information such as textures, colors, key points, and fine-level features. This abstraction exacerbates the inherent ill-posedness and camera-shape ambiguity in single-view 3D human reconstruction Li et al. (2020). Although significant progress has been made in reconstructing 3D models from 2D images Pontes et al. (2019); Wang et al. (2018); Kolotouros et al. (2019a); Kanazawa et al. (2018); Kolotouros et al. (2019b), existing methods Luo et al. (2021); Brodt & Bessmeltsev (2022) encounter substantial challenges when applied to sketches due to their sparse and abstract characteristics. The most related work, SketchBodyNet Wang et al. (2023), addresses sketch-based 3D reconstruction but is constrained in its ability to represent diverse body shapes, largely due to limited data availability. While there are abundant RGB-based 3D human datasets, well-paired sketch-mesh datasets remain scarce.

To address these limitations, we first introduce HybridSketchNet, which exploits 2D point clouds extracted from sketches as an additional data modality for body shape modeling. As illustrated in Figure 1, our method provides an overview of the proposed network and showcases several reconstruction examples from hand-drawn sketches. Specifically, HybridSketchNet employs a hybrid

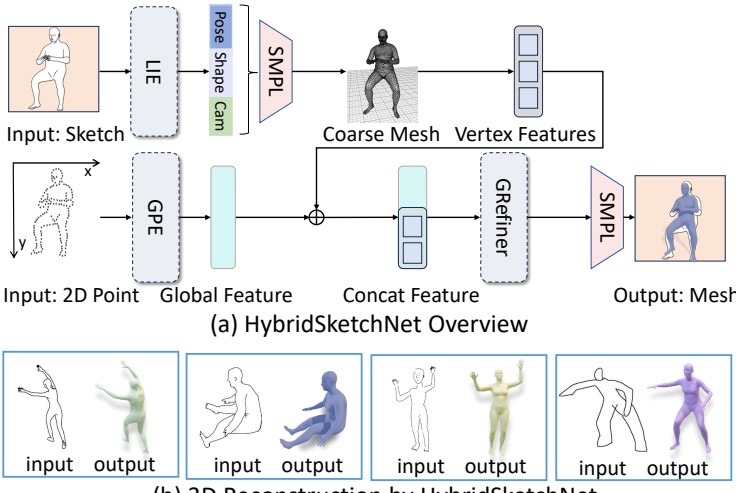

(a) HybridSketchNet Overview

(b) 3D Reconstruction by HybridSketchNet

Figure 1: Overview of proposed network (a) and four reconstructed mesh examples from hand-drawn sketches (b). The aim is to generate a three-dimensional human body mesh that precisely corresponds to sketches illustrating different body shapes.

architecture, using geometric characteristics to enhance image features through three modules: (1) a transformer-based local image encoder (LIE) to process sketch images, which captures reliable pose information; (2) a global point encoder (GPE) that extracts semantic and geometric features from the 2D point clouds of sketches; (3) a graph-based refiner (GRefiner) that fuses both features through cross-modal graph-based integration, ultimately enabling the reconstruction of 3D human meshes with accurate poses and shapes. To further address the lack of suitable datasets for sketch-to-3D human reconstruction, we developed Sketch3DS, a large-scale dataset comprising nearly 10,000 synthesized sketch–3D mesh pairs and 10,000 real pairs with hand-drawn sketches. The synthetic portion combines a wide range of postures from existing 3D human datasets with randomly generated body shapes, effectively covering diverse postures and body types observed in real-world scenarios. This synthesized portion facilitates pretraining of a reconstruction model that disentangles pose and shape. The model is subsequently fine-tuned on the Sketch3DS dataset, thereby advancing the task of 3D human reconstruction from sketches.

In summary, our contributions are as follows:

- We propose HybridSketchNet, a hybrid framework exploiting both local and global information from sketches and 2D point clouds to estimate SMPL parameters in a coarse-to-fine manner.

- We present Sketch3DS, a large-scale dataset encompassing various body shapes and poses, facilitating research on sketch-based body shape reconstruction.

- Extensive experimental results demonstrate that HybridSketchNet outperforms existing methods significantly.

## 2 RELATED WORK

**Single-view 3D Human Reconstruction.** Traditional methods for three-dimensional reconstruction from monocular images require at least two images for calculating the three-dimensional coordinates and invariably involve camera calibration to ascertain both intrinsic and extrinsic parameters for object transformations. However, sketches lack such camera information, rendering many algorithms based on camera calibration ineffective. Recent advancements in deep neural networks have led to significant progress in regressing 3D joints and body shapes from 2D keypointsChen et al. (2022a;b); He et al. (2016); Wu et al. (2019). These methods establish a direct link between two-dimensional data and three-dimensional models, emulating camera calibration transformations and offering inno-

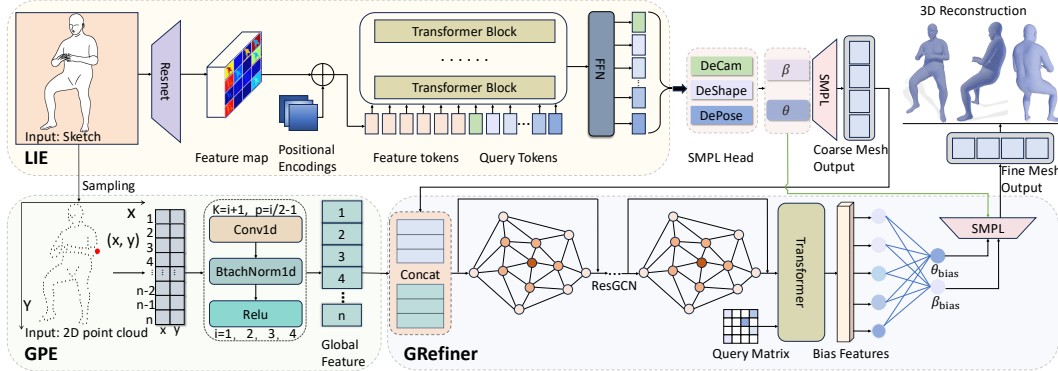

Figure 2: Overview of HybridSketchNet. The network includes three modules: the Local Image Encoder (LIE), the Global Point Encoder module (GPE), and the Graph-based Refiner module (GRefiner). It aims to generate a three-dimensional human body that accurately meshes with sketches that reflect changes in human body shape.

vative approaches for reconstructing three-dimensional models from single images. Notably, Martinez et al. Martinez et al. (2017) transformed 2D keypoints into 3D joints, while Choi et al. Choi et al. (2020) introduced PoseNet and Mesh-Net for direct mesh prediction. PoseFormer Oreshkin (2023) adapted Transformer architecture for pose transformation.

End-to-end SMPL-based methods Loper et al. (2015) streamline computational processes. Kanazawa et al. Varol et al. (2018) developed models for SMPL parameter prediction. Kolotouros et al. Kolotouros et al. (2019a) employed SMPLify for self-monitoring networks, with subsequent improvements through synthetic training data Sengupta et al. (2020), image heatmaps Moon & Lee (2020), multi-cycle prediction Zhang et al. (2021a), differentiable semantic loss Dwivedi et al. (2021), multilevel attention Wan et al. (2021), and hybrid inverse kinematics Li et al. (2021). However, applying these methods to sketches remains challenging due to their sparse nature.

**3D Reconstruction Based on Sketches.** Sketches serve as efficient tools in applications including inpainting Gao et al. (2017); Qi et al. (2021), synthesis Li et al. (2017), retrieval Deng et al. (2018); Eitz et al. (2010), segmentation Wang et al. (2020a); Zheng et al. (2024), scene generation Shin & Igarashi (2007), structural analysis Yu et al. (2023), and shape retrieval Wang et al. (2017; 2016).

Recent deep learning approaches advanced sketch-based 3D reconstruction: Wang et al. Wang et al. (2020b) reconstructed point clouds, Luo et al. Luo et al. (2021) focused on animal reconstruction, Zhang et al. Zhang et al. (2021b) addressed sketch ambiguity, and Wu et al. Wu et al. (2023) proposed diffusion models. For human body reconstruction, Unlu et al. Unlu et al. (2022) modeled body parts as cylinders, while Brodt and Bessmeltsev Brodt & Bessmeltsev (2022) used skeletal tangents and foreshortening. Key challenges remain: (1) dataset scarcity, and (2) sketch randomness, causing overfitting.

## 3 HYBRIDSKETCHNET

As shown in Figure 2, HybridSketchNet adopts a hybrid framework with three core modules: (1) Local Image Encoder (LIE) using ViT architecture to process sketch images and predict coarse SMPL parameters; (2) Global Point Encoder (GPE) using PointNet to extract global semantic features from 2D point clouds sampled from sketches; (3) Graph-based Refiner (GRefiner) that fuses local image features and global point cloud features to predict parameter offsets for final mesh refinement. This coarse-to-fine approach leverages complementary information from both sketch images and point clouds.

## 3.1 PROBLEM DEFINITION

Given a human sketch, our task is to reconstruct its corresponding mesh by fitting a parametric human model. Particularly, we utilize the widely-used SMPL model Loper et al. (2015) in this paper, which represents human meshes by a predefined human mesh template $\overline{T} \in \mathcal{R}^{6890 \times 3}$ and vertex deformations governed by a set of pose and shape parameters $(\theta, \beta)$. The pose parameter $\theta \in \mathbb{R}^{72}$ represents the 3D rotation angles of 24 human body joints, and the shape parameter $\beta \in \mathbb{R}^{10}$ is the coefficients of the principal components learned on various human bodies.

## 3.2 LOCAL IMAGE ENCODER

Processing sparse sketch lines with traditional CNNs is ineffective as they focus on local structures rather than long-range information. Therefore, we adopt a ViT-based architecture in LIE to better exploit and aggregate local features from sketches.

**ViT Encoder.** Input sketch $I \in \mathbb{R}^{H \times W}$ is processed by ResNet50 encoder to obtain feature maps $\mathbf{x} \in \mathbb{R}^{\frac{H}{s} \times \frac{W}{s} \times C}$ where $C = 2048$ and $s = 8$. Since spatial information helps depict poses, we employ positional encoding to incorporate spatial information:

$$\mathbf{x_0} = \mathbf{x} + \mathcal{PE}(H_0, W_0) \tag{1}$$

where $\mathcal{PE}(H_0, W_0) \in \mathbb{R}^{2048 \times H_0 \times W_0}$ represents learnable position embeddings. Features are flattened into tokens and processed by Transformer encoder with learnable queries $\mathcal{E}_Q \in \mathbb{R}^{30 \times 2048}$. The Transformer uses self-attention and cross-attention mechanisms to refine understanding of human posture from image features.

The Transformer encoder, inspired by DETR Carion et al. (2020), processes learnable queries $\mathcal{E}_Q \in \mathbb{R}^{30 \times 2048}$ and image features $\mathbf{x_2} \in \mathbb{R}^{L \times 2048}$ through self-attention and cross-attention:

$$\S_{out} = \mathcal{F}_{softmax} \left( \frac{q_x \cdot k_x^T}{\sqrt{d_x}} \right) v_x. \tag{2}$$

The output is transformed through the projection layer into $\mathcal{E}_f \in \mathbb{R}^{30 \times 2048}$ for the SMPL Head.

**SMPL Head** contains three MLP decoders that receive Transformer output $\mathcal{E}_f$ and predict initial parameters:

$$\theta = DePose(\mathcal{E}_{f_1}), \quad \beta = DeShape(\bar{\mathcal{E}}_{f_2})$$
$$\mathcal{C}am = DeCam(\mathcal{E}_{f_3}) \tag{3}$$

For shape reconstruction, we employ averaging to aggregate body shape features into global feature $\bar{\mathcal{E}}_{f_2}$. The parameters $\theta$ and $\beta$ generate coarse mesh $M_{coarse} \in \mathbb{R}^{n \times 3}$ for subsequent refinement.

## 3.3 GLOBAL POINT ENCODER

PointNet Qi et al. (2016) is notable for its indifference to point order, robustness against perturbations, and ability to handle variable point numbers, making it effective for 2D point cloud processing. We uniformly sample 512 coordinates from sketch contours to generate point cloud data that abstractly represents body posture and shape features.

The GPE consists of four PointBlocks with output channels [64, 64, 128, 256]. Each PointBlock contains one-dimensional convolution, normalization, and activation layers. Point clouds pass through these PointBlocks to derive hierarchical features, followed by max-pooling to obtain global vector $\mathbf{y_3} \in \mathbb{R}^{256 \times 1}$ encapsulating all contextual information. This global feature is replicated to create $\mathbf{y} \in \mathbb{R}^{256 \times n}$ for each mesh vertex, where $n$ is the number of vertices in the coarse mesh $M_{coarse}$ from LIE.

## 3.4 GRAPH-BASED REFINER

To enhance the accuracy of 3D human mesh reconstruction and improve responsiveness to changes in human body shape, we developed the Graph-based Refiner module (GRefiner). This module utilizes a combination of GraphCNN Kolotouros et al. (2019b), Transformer, and MLP to fine-tune the predicted SMPL parameters for effective reconstruction of sketched human body meshes.

**ResGCN** module consists of graph linear transformation layers and multiple GraphResBlocks with residual connections. Global point cloud features $\mathbf{y} \in \mathbb{R}^{256 \times n}$ are transposed to $\mathbf{y^T} \in \mathbb{R}^{n \times 256}$ and concatenated with coarse mesh: $F_{vertex} = [M_{coarse}, \mathbf{y^T}]$. This ensures each vertex incorporates global point cloud features, resulting in enhanced features $F_{vertex} \in \mathbb{R}^{n \times 259}$ containing 3D pose and 2D shape information. The enhanced features are processed through Graph Neural Network (GCN) Kolotouros et al. (2019b) according to:

$$\mathbf{Y} = \tilde{\mathbf{A}} \mathbf{F_I} \mathbf{W}, \tag{4}$$

where $\mathbf{F_I} \in \mathbb{R}^{n \times D_{in}}$ represents the intermediate feature matrix, $\mathbf{W} \in \mathbb{R}^{D_{in} \times D_{out}}$ is the weight parameter matrix, and $\tilde{\mathbf{A}} \in \mathbb{R}^{n \times n}$ denotes the adjacency matrix. The output $\mathbf{Y} \in \mathbb{R}^{256 \times n}$ is processed through two branches: (1) a GCN decoder that directly predicts vertex position offsets $M_{bias} \in \mathbb{R}^{n \times 3}$; (2) an MLP that produces abstract mesh features $F \in \mathbb{R}^{49 \times 256}$ for SMPL parameter prediction.

**Offset Regressor** A Transformer-based regressor processes abstract mesh features $F \in \mathbb{R}^{49 \times 256}$ with positional embeddings $\mathcal{E}_R \in \mathbb{R}^{34 \times 256}$ to produce offset features $\bar{F} \in \mathbb{R}^{34 \times 256}$. The feature $\bar{F}$ is divided into pose and shape components: $\bar{F}_p \in \mathbb{R}^{24 \times 256}$ for pose adjustments and $\bar{F}_s \in \mathbb{R}^{10 \times 256}$ for shape adjustments. These are processed by separate MLPs:

$$\beta_{bias} = DeShapeBias\left(\bar{F}_s\right), \tag{5}$$

$$\theta_{bias} = DePoseBias\left(\bar{F}_p\right). \tag{6}$$

where $\beta_{bias} \in \mathbb{R}^{10}$ and $\theta_{bias} \in \mathbb{R}^{72}$ are the shape and pose parameter offsets, respectively. The final refined mesh is generated as $M_{final} = SMPL(\theta + \theta_{bias}, \beta + \beta_{bias})$.

### 3.5 LOSS FUNCTION

We use HuberLoss Huber (1992) for robustness against outliers in sketch data:

$$\text{HuberLoss}(y, \hat{y}) = \begin{cases} \frac{1}{2}(y - \hat{y})^2, & \text{if } |y - \hat{y}| \leq \delta \\ \delta(|y - \hat{y}| - \frac{1}{2}\delta), & \text{otherwise,} \end{cases} \tag{7}$$

where $\delta$ determines the threshold between L1 and L2 loss. This reduces the impact of outliers during training.

The loss combines multiple components: 3D joint loss $\mathcal{L}_{V_{3D}}$, 2D joint loss $\mathcal{L}_{V_{2D}}$, SMPL parameter losses $\mathcal{L}_\beta$ and $\mathcal{L}_\theta$, and offset losses $\mathcal{L}_{\beta_{bias}}$ and $\mathcal{L}_{\theta_{bias}}$. We also supervise mesh vertex losses for different stages: $M_{coarse}$, $M_{bias}$, $(M_{coarse} + M_{bias})$, and $M_{final}$, aggregated as:

$$\mathcal{L}_{sum} = \sum_{i=1}^{4} \mathcal{L}_i \times W_i. \tag{8}$$

The final loss function is expressed as follows:

$$\begin{aligned} \mathcal{L} = & \mathcal{L}_{V_{3D}} \times W_{V_{3D}} + \mathcal{L}_{V_{2D}} \times W_{V_{2D}} + \mathcal{L}_\beta \times W_\beta \\ & + \mathcal{L}_\theta \times W_\theta + \mathcal{L}_{\beta_{bias}} \times W_{\beta_{bias}} \\ & + \mathcal{L}_{\theta_{bias}} \times W_{\theta_{bias}} + \mathcal{L}_{sum}, \end{aligned} \tag{9}$$

where $W$ is a hyperparameter that controls the relative weight of each loss component.

## 4 SKETCH3DS DATASET

We present Sketch3DS, a large-scale synthetic dataset comprising 10,000 pairs of SMPL parameters, designed to generate sketches that capture a wide variety of human postures and body shapes. Drawing inspiration from the scale and diversity of Human3.6M Ionescu et al. (2013), Sketch3DS provides a comprehensive resource for research in human modeling and related tasks. We randomly select 10,000 images for sketching tasks to establish the sketch-rendered human body image data pairs.

**Data Collection.** We extracted over 10,000 projection images from our Sketch3DS dataset and organized them into 34 groups, each containing 300 images. Thirty-four student volunteers from

| Category | Min | Max | Mean | Std |
|---|---|---|---|---|
| Wang et al. Wang et al. (2023) | | | | |
| Total | 233 | 39953 | 8383 | 6134 |
| Train | 233 | 39953 | 8349 | 6132 |
| Valid | 1688 | 31410 | 8641 | 6143 |
| Sketch3DS (ours) | | | | |
| Total | 1009 | 143945 | 14880 | 8203 |
| Train | 1009 | 143945 | 14867 | 8205 |
| Valid | 1200 | 107874 | 14993 | 8185 |

Table 1: Diversity statistics comparisons for our Sketch3DS dataset against the pioneering one Wang et al. (2023).

diverse academic backgrounds participated in the sketching process. Unlike the protocol adopted in the pioneering work Wang et al. (2023), our approach imposed no strict requirements on participant qualifications or the electronic devices used for drawing. After data collection, we compiled nearly 10,000 valid hand-drawn sketches. These validated sketches were subsequently divided into training and validation sets in a 9:1 ratio. Our Sketch3DS includes diverse poses and body shapes depicted in various drawing styles.

The absence of strict constraints on volunteers' drawing abilities and equipment resulted in a wide range of sketching styles, thereby significantly enhancing the diversity of the dataset. Individual drawing habits and tool choices affected the number of effective data points (i.e., pixels representing the human figure, excluding the background) per sketch. Table 1 presents a statistical analysis of this diversity, reporting the minimum, maximum, mean, and standard deviation of effective data points, thereby confirming the substantial heterogeneity within our dataset.

**Data Preprocessing.** We localized human figures using pixel statistics to determine center points and cropping lengths, saving parameters ($center$, $length$, $\theta$, $\beta$) and resizing to $256 \times 256$ pixels. Specifically, we computed statistics on image pixels to determine the maximal and minimal positions containing pixels, and then deduced the center point $center$ and cropping length $length$ of the human figure. Subsequently, we saved the corresponding $center$, $length$, SMPL poses $\theta$, and SMPL shapes $\beta$ in a list, which was stored as a numpy file for easy loading during training and prediction phases. Additionally, we resized the images to $256 \times 256$ pixels. Data augmentation included rotation, scaling, and noise, with point clouds receiving the same transformations and centroid normalization. Given the variable resolutions of the sketches, which led to inconsistencies in the time complexity of the rotation algorithm, we standardized the process by cropping and converting images to $512 \times 512$ resolution before rotating them. This adjustment significantly reduced training times once rotation was enabled. To ensure that point cloud data received the same rigid transformations, we used previously saved cropping parameters for cropping and rigid transformations, then extracted the point cloud coordinates from the sketches. Moreover, for better feature extraction from the point clouds, we normalized the point clouds to their centroids. It should be noted that during the training phase, we adopted a random dropout strategy for data augmentation.

## 5 EXPERIMENTS

### 5.1 IMPLEMENTATION DETAILS

We pre-trained the model on rendered human body images (learning rate 4e-5), then fine-tuned it on sketches (learning rate 1e-5). This transfer learning approach enhanced performance. We trained on both Sketch3D and Sketch3DS datasets, computing MPJPE on Sketch3D and both MPJPE and MPVPE on Sketch3DS. Data augmentation included noise (0.4), scaling (0.25), and rotation (-30° to 30°) for both images and point clouds.

| Model | Sketch3D | | Sketch3DS | | | |
|---|---|---|---|---|---|---|
| | Synthesis | Freehand | Synthesis | | Freehand | |
| | MPJPE↓ | MPJPE↓ | MPJPE↓ | MPVPE↓ | MPJPE↓ | MPVPE↓ |
| HMR Kanazawa et al. (2018) | 158 | 180 | 110 | 133 | 164 | 203 |
| MeshPose* Le et al. (2024) | 351 | 376 | 238 | 258 | 317 | 338 |
| Sketch2Pose* Brodt & Bessmeltsev (2022) | 259 | 313 | 219 | 244 | 288 | 301 |
| SPIN Kolotouros et al. (2019a) | 123 | 161 | 82 | 104 | 135 | 173 |
| CMR Kolotouros et al. (2019b) | 128 | 163 | 67 | 79 | 138 | 172 |
| MAED Wan et al. (2021) | 119 | 146 | 59 | 73 | 117 | 148 |
| Hybrik Li et al. (2021) | 124 | 155 | 54 | 69 | 116 | 148 |
| SketchBodyNet Wang et al. (2023) | 127 | 155 | 64 | 80 | 125 | 159 |
| Ours | **114** | **139** | **49** | **63** | **99** | **124** |

Table 2: Compared with the state-of-the-art methods on Sketch3D, we only evaluated the MPJPE as we lacked shape information. Our method is superior to the previous methods in all metrics for both freehand and synthetic data. * indicates methods without available training code.

## 5.2 EVALUATION METRICS

To evaluate the performance of our body shape reconstruction, we introduced the Mean Per-Vertex Position Error (MPVPE), which is a commonly used measure to assess the accuracy of mesh estimation in 3D reconstruction. A lower MPVPE value indicates that the reconstruction result is closer to the ground truth mesh, indicating a more accurate reconstruction. It calculates the mean square error of the human body model. The calculation formula is as follows:

$$E_{MPVPE}(f, \mathcal{V}) = \frac{1}{N_{\mathcal{V}}} \sum_{i=1}^{N_{\mathcal{V}}} \|m_{\mathbf{f},\mathcal{V}}^{(f)}(i) - m_{\mathbf{gt},\mathcal{V}}^{(f)}(i)\|_2. \tag{10}$$

Where $\mathcal{V} \in \mathbb{R}^{6890 \times 3}$ represents the vertices of the three-dimensional human body model generated by SMPL.

Following Wang et al. (2023), we use the Mean Per-Joint Position Error (MPJPE) Ionescu et al. (2013) as our evaluation metric, following its definition:

$$E_{MPJPE}(f, S) = \frac{1}{N_S} \sum_{i=1}^{N_S} |m_{f,S}^{(f)}(i) - m_{gt,S}^{(f)}(i)|_2. \tag{11}$$

## 5.3 COMPARE TO STATE-OF-THE-ART METHODS

We compared our network with several typical SMPL-based methods from 2018 to 2023, training all models on our synthetic and sketch datasets with the same learning rate. The results in Tables 2 show our network achieves the best performance, outperforming Hybrik by 14% on the Sketch3DS dataset. Compared to previous methods, our approach maps the predictions of body shape and pose separately to the image and point cloud domains. By leveraging the relationship between the 3D model's boundary points and the 2D points in the sketch, our method further enhances the prediction of body shape. Therefore, our method has a distinct advantage over previous approaches in predicting both body shape and pose accurately.

From the comparative experiment results, it is evident that our network outperforms existing methods in terms of both pose and body shape reconstruction on both synthetic and hand-drawn sketches. As shown in Table 2, our method surpasses previous approaches, even when trained on Sketch3D datasets with only pose labels. This improvement can be attributed to the GPE (Global Point Encoder) module, GRefiner (Graph-based Refiner) module and LIE (Local Image Encoder) module. This module utilizes the basic image features and offsets features of 2D points to predict the pose parameters of SMPL. This method of combining basic image features and offset features can further improve the accuracy of reconstruction.

Furthermore, as indicated in Table 2, our method continues to exhibit superior performance when applied to Sketch3DS datasets that include body shape information. Compared to the excellent de-

| MODEL | Part | | | Sketch3D | | Sketch3DS | | | |
|---|---|---|---|---|---|---|---|---|---|
| | LIE | GPE | GCN | Synthesis | Freehand | Synthesis | | Freehand | |
| | | | | MPJPE↓ | MPJPE↓ | MPJPE↓ | MPVPE↓ | MPJPE↓ | MPVPE↓ |
| GCN-to-MLP | ✓ | ✓ | | 115 | 142 | 52 | 66 | 101 | 127 |
| LIE-Only | ✓ | | | 118 | 140 | 53 | 68 | 104 | 131 |
| GPER | | ✓ | ✓ | 194 | 244 | 175 | 212 | 213 | 265 |
| Ours | ✓ | ✓ | ✓ | **114** | **139** | **49** | **63** | **99** | **124** |

Table 3: Ablation experiments on Sketch3D dataset. We only calculate the MPJPE metric on the Sketch3D dataset. At the same time, the ablation experimental results on the Sketch3DS dataset are put together for comparison.

coupled structure of Hybrik, our method demonstrates significant improvement in both pose and body shape. This is attributed to our method's ability to decouple the prediction of SMPL parameters, incorporate offset position encoding to identify pose and body shape vectors, and employ attention mechanisms for hierarchical selection on these offset vectors. By simple mapping, we can predict the offset values of $\theta$ and $\beta$ parameters of SMPL from 2D point coordinates. Additionally, we introduced three specific loss functions to supervise the offset of $\theta$, $\beta$, and mesh coordinates. Experimental results have demonstrated that combining the offset features of the GRefiner module with special position encoding significantly enhances the reconstruction performance of the network.

## 5.4 ABLATION STUDY

In our ablation studies, we implemented the following modifications to evaluate the contributions of different components in HybridSketchNet using three distinct network architectures: (1) **GCN-to-MLP**: By removing the Graph Neural Networks from the GRefiner modules and replacing them with MLPs, we assessed the critical role of the Graph Neural Networks in HybridSketchNet. (2) **LIE-Only**: By omitting the entire GPE and GRefiner modules and retaining only the LIE module to extract sketch features for coarse 3D human body reconstruction, we evaluated the impact of this module on the network. (3) **GPER**: We used the GPE and GRefiner modules alone to perform 3D reconstruction using 2D point cloud data.

The results in Tables 3 highlight the pivotal role of GCNs in fusing features from 2D point clouds and 3D meshes. Replacing GCNs with standard MLPs leads to significant drops in pose and shape accuracy, as MLPs lack the graph-based constraints that keep predictions within the valid SMPL space, often resulting in unrealistic outputs. Additionally, comparisons between LIE-Only and GCN-to-MLP models show that point clouds alone have limited impact on pose regression, with only slight improvements observed in datasets without body shape variation. Removing the GPE and GRefiner modules and relying solely on CNN and Transformer-based regression yields performance similar to prior methods, but incorporating these modules to refine the mesh using point cloud features further improves reconstruction accuracy. Ablation results also demonstrate that predicting SMPL parameters directly from point sets is not feasible, as the geometric information in 2D point clouds is too abstract for simple networks. Therefore, integrating GPE and GRefiner as offset components within our network is essential for achieving superior 3D reconstruction performance.

## 5.5 USER STUDY

We have conducted a user study to assess the fidelity of the 3D models generated by the network to the input sketches, as well as the visual quality of pose and body shape reconstructions. A set of 50 input sketches was prepared, with each sketch accompanied by nine files: the sketch source file, the corresponding ground truth (GT) file, and 3D mesh model files generated by seven different neural network models. In this study, we recruited another 31 college students (15 males and 16 females) ranging in age from 20 and 26, with academic backgrounds in fields such as computer science and electronic information. We first created a scoring guideline document to clarify the tasks and evaluation criteria for each volunteer. Volunteers were instructed to rate the models based on the following criteria: (1) the faithfulness of each model to the sketch source file, (2) the visual accuracy of pose reconstruction compared to the GT file, and (3) the visual accuracy of body shape

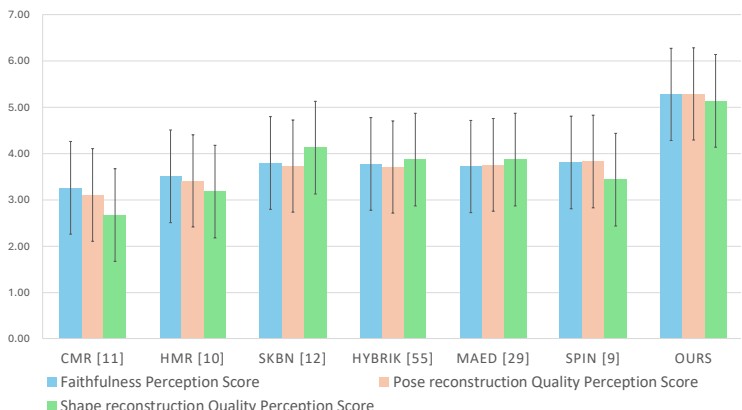

Figure 3: The average quality and faithfulness perception scores for the methods selected for comparison.

reconstruction compared to the GT file. The volunteers rated each result using a seven-point Likert scale (1 = strongly negative to 7 = strongly positive).

Overall, we gathered 31,920 subjective evaluations (50 sketches × 31 volunteers × 3 criteria × 7 models). To ensure fairness, we anonymized each file according to a uniform naming convention so that volunteers would not know which model each file corresponded to.

As shown in Figure 3, our method received high scores in faithfulness ($F = 97.65, p < 0.001$), pose reconstruction ($F = 64.12, p < 0.001$), and body shape reconstruction ($F = 61.94, p < 0.001$). In terms of faithfulness, our proposed HybridSketchNet model achieved the highest perceived score (5.28 ± 1.07) compared to the other six methods: CMR (3.51 ± 1.32), HMR (3.51 ± 1.20), Sketch-BodyNet (3.80 ± 1.17), Hybrik (3.78 ± 1.14), MAED (3.72 ± 1.32), and SPIN (3.81 ± 1.13). Regarding pose reconstruction, our method (5.29 ± 1.12) also outperformed CMR (3.11 ± 1.40), HMR (3.41 ± 1.11), SketchBodyNet (3.73 ± 1.18), Hybrik (3.71 ± 1.13), MAED (3.76 ± 1.22), and SPIN (3.83 ± 1.18). As for body shape reconstruction, volunteers rated our method (5.14 ± 1.11) as the best, compared to CMR (2.67 ± 1.44), HMR (3.19 ± 1.29), SketchBodyNet (4.14 ± 1.15), Hybrik (3.86 ± 1.20), MAED (3.88 ± 1.52), and SPIN (3.44 ± 1.23). These results further demonstrate the superiority of our approach.

## 6 CONCLUSION

In this paper, we propose HybridSketchNet, a hybrid framework for 3D human reconstruction from sketches. Our framework integrates sketch features with 2D point cloud data to enable both pose and body shape reconstruction directly from sketches. Comparative experiments on two sketch datasets demonstrate superior performance over state-of-the-art methods. The use of HuberLoss helps address outlier issues in sketch datasets.

In the future, improving the performance of the model can be achieved by increasing the diversity and accuracy of the dataset. This can be done by collecting more varied and precise sketch data. By continuing to enhance the dataset and exploring novel techniques, we can further improve the model's performance in accurately reconstructing 3D models from sketches.

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
