# OpenReview forum: "HybridSketchNet: Sketch-based 3D Human Mesh Reconstruction via Hybrid Point-Image Networks"
_ICLR.cc/2026/Conference — Submitted to ICLR 2026_

### Official Review · Reviewer_G47j · 2025-10-15

**Soundness:** 2
**Presentation:** 2
**Contribution:** 2
**Rating:** 2
**Confidence:** 5

**Summary:**

This paper introduces HybridSketchNet, a framework for reconstructing 3D human meshes from hand-drawn sketches. The method leverages three components: (1) a Local Image Encoder (LIE) based on ViT for coarse pose estimation, (2) a Global Point Encoder (GPE) using sampled sketch contours for shape features, and (3) a graph-based refiner (GRefiner) that integrates local and global features for mesh refinement. To support this, the authors also build Sketch3DS, a new dataset of ~10,000 paired sketches and 3D human meshes, combining synthetic renderings and real hand-drawn sketches. Experiments show that the proposed method outperforms existing baselines on sketch-based 3D reconstruction tasks.

**Strengths:**

1. Novel problem definition: The paper addresses sketch-based 3D human reconstruction, which is less explored compared to image-based methods.

2. Dataset contribution: The authors introduce Sketch3DS, a large-scale paired sketch-mesh dataset, which could serve as a resource for future research.

**Weaknesses:**

1. Questionable motivation for dataset creation: The necessity of a new sketch-mesh paired dataset is unclear. An alternative pipeline could be sketch → image generation → existing image-based 3D human pose/mesh estimation, which would leverage abundant image datasets and avoid the inefficiency of building a new paired dataset.

2. Over-engineering of refinement: The proposed GRefiner uses a graph-based architecture, but the gains may simply come from additional parameters rather than the graph design itself. A simpler refinement module (e.g., MLP layers on top of concatenated features, or a transformer-only refinement) might yield similar improvements.

3. Narrow evaluation scope: While the proposed method improves on sketch benchmarks, the paper does not convincingly argue why sketch-to-3D reconstruction is preferable or necessary compared to sketch-to-image plus existing pipelines.

**Questions:**

Please refer to Weaknesses.

---

### Official Review · Reviewer_72A2 · 2025-10-26

**Soundness:** 3
**Presentation:** 2
**Contribution:** 3
**Rating:** 4
**Confidence:** 3

**Summary:**

This paper presents HybridSketchNet for reconstructing 3D human meshes from hand-drawn sketches. The system has three components: a Local Image Encoder using ResNet50 plus Transformer to get initial SMPL parameters and a coarse mesh, a Global Point Encoder based on PointNet that processes 512 sampled contour points, and a Graph-based Refiner that combines features from both streams using GraphCNN, Transformer, and MLP layers to predict vertex and parameter offsets. The authors introduce Sketch3DS with nearly 10k hand-drawn sketches and report improvements in MPJPE and MPVPE over existing methods on both Sketch3D and Sketch3DS. A user study with 31 participants shows preference for their outputs.

**Strengths:**

The work addresses sketch-based 3D human reconstruction, which has received less attention than image or video-based methods. The main contributions are the hybrid architecture that processes both pixel-level and contour-level representations, and the Sketch3DS dataset with roughly 10k real hand-drawn sketches. The experimental results show measurable improvements.

The technical design is reasonable, and the experiments support the claims. The paper includes proper baselines on Sketch3D and Sketch3DS, and the ablations demonstrate that each component matters. The LIE-Only variant performs similarly to prior image-based methods, the GPER variant that predicts SMPL directly from points performs poorly, and replacing GCN with MLP causes significant drops. The training procedure of pretraining on rendered images then finetuning on sketches is sensible.

The hybrid design makes sense for sketches where local details indicate pose and global contours indicate shape. The ablations validate this choice by showing that image-only, points-only, and versions without graph structure all perform worse.
The dataset of nearly 10k hand-drawn sketches fills a gap. The paper describes the collection process and uses synchronized augmentation for both image and point inputs during training. Pretraining on synthetic data before finetuning on real sketches is a practical strategy.

The evaluation includes quantitative metrics on two datasets, comparisons with multiple baselines including SPIN, HybrIK, and SketchBodyNet, and a user study with 31 participants using anonymized outputs and Likert scale ratings.

**Weaknesses:**

The approach combines existing building blocks rather than introducing new architectural primitives or learning techniques. The dataset is useful but has acknowledged limitations in quality and diversity that the authors mention needing to address. This is solid applied work on a practical problem rather than a fundamental advance.

The camera model is unspecified. Sketches do not contain perspective cues that photos have. How does the model handle camera parameters? Is there a separate camera prediction branch? Are camera parameters fixed? How is this supervised?

PA-MPJPE would show pose accuracy independent of global alignment. Per-vertex error would show whether improvements are uniform across the body or concentrated in certain regions. Without these it is a bit harder to assess what exactly the model learns.
The dataset has limitations the authors acknowledge. The real sketches come from volunteers with varying drawing skill and no equipment standardization, creating heterogeneity in style and quality. The paper mentions relying on synthetic data for pretraining but does not quantify the domain gap or show cross-domain results like training on synthetic and testing on real sketches. More visual examples and error analysis is needed.

The architecture has strong dependencies. The points-only variant (GPER) fails badly, showing you cannot predict SMPL from 2D contours alone. The GCN-to-MLP ablation shows large drops, indicating the graph structure is load-bearing. This suggests limited modularity and that performance depends on the specific combination of components.

Reproducibility details are incomplete: optimizer, learning rates, batch sizes, number of training epochs, and augmentation parameters should be specified. There is no discussion or visualization of failure cases. No analysis of what happens with stylized sketches, occlusions, or extreme foreshortening.

Regarding presentation:
The paper is generally well-structured. The pipeline description covers the main steps. Figures could be improved with more visual examples. Limitations of this method with sketch examples and error analysis are lacking. The user study figure is somewhat informative, but the statistical analysis and the figure could be presented in a table.
The graph connectivity structure used in GRefiner could be described more precisely. Training hyperparameters like batch size, learning rate schedule, and number of epochs are incomplete.
Computer Graphics is another domain which studied sketches in depth and some related work overview could be considered.

**Questions:**

See Weaknesses

---

### Official Review · Reviewer_Jm78 · 2025-10-29

**Soundness:** 3
**Presentation:** 2
**Contribution:** 2
**Rating:** 4
**Confidence:** 4

**Summary:**

The authors tackle the problem of reconstructing 3D human meshes from 2D sketches. They note two main challenges: sketches are abstract and sparse compared to natural images, and existing datasets of sketch-to-mesh pairs are lacking. To address these, they propose a model includes a Local Image Encoder (LIE) (ViT) that processes the sketch image to estimate a coarse human mesh, a Global Point Encoder (GPE) that takes 2D contour coordinates from the sketch (points) to obtain a global representation focusing on body shape and a Graph‐based Refiner (GRefiner) that fuses the local and global representations and outputs a finely fitted mesh. The authors also created a dataset including10,000 sketch‐mesh pairs with diverse body shapes and poses. Experiments show that their method outperforms existing approaches on this dataset and yields accurate alignment between input sketch and output mesh.

**Strengths:**

- The paper identifies an interesting problem (sketch to 3D human mesh) and proposes a workable solution. The architecture is technically sound, with pixel and point based encoding combined with GNN.
- The introduction of a dedicated dataset (Sketch3DS) is valuable for future work. The empirical improvement reported adds credibility to the proposed architecture.

**Weaknesses:**

- The presentation can be improved, especially the numbers in the Ablation Study.
- The visual results in this paper is very limited, making it hard to judge the quality of the dataset and final results.
- It would be interesting if the authors can provide the analysis of the performance gain over the dataset size.

**Questions:**

- What is the motivation of using GNN? How would that compared to transformers?

---

### Official Review · Reviewer_wFr2 · 2025-10-29

**Soundness:** 2
**Presentation:** 3
**Contribution:** 2
**Rating:** 4
**Confidence:** 4

**Summary:**

This paper proposed a sketch-based 3D human mesh reconstruction method. 3D human reconstruction from sketches is interesting. The experimental results demonstrate the effectiveness of the proposed method.

**Strengths:**

（1） The construction and release of the large-scale Sketch3DS dataset provides a valuable resource for the community and helps alleviate the critical problem of data scarcity in sketch-based 3D human reconstruction.
（2）The proposed HybridSketchNet framework explores a multi-modal approach by fusing sketch images and 2D point clouds. This is a meaningful direction for tackling the abstract and sparse nature of sketch data.
（3） The paper provides a multi-faceted evaluation, including quantitative comparisons, ablation studies, and a user study.

**Weaknesses:**

（1）The primary baselines for comparison (e.g., HMR, CMR, SPIN, HybrIK, MAED) are models designed for 3D reconstruction from natural images. These models rely heavily on textures, lighting, and contextual cues abundant in RGB photos. Applying them directly to information-poor sketches creates a significant domain mismatch, artificially depressing their performance. This setup unfairly amplifies the perceived advantage of the proposed method. A rigorous and convincing comparison should primarily focus on and demonstrate clear superiority against state-of-the-art methods specifically designed for sketches (like SketchBodyNet). The current comparisons fail to cleanly isolate the true contribution of the proposed architecture in advancing sketch understanding itself, rather than just showcasing the expected advantage of a domain-specific model over general-purpose ones.
（2） While SketchBodyNet is included, the comparison against it—now the most critical benchmark given the baseline issue—lacks depth. A more thorough analysis is needed to explain why and how the hybrid architecture outperforms SketchBodyNet on its own turf.
（3）The "hand-drawn sketches" presented in the paper and within the Sketch3DS dataset appear stylistically uniform, with clean and continuous lines, closely resembling extracted contours from 3D model renderings. They do not reflect the highly abstract, ambiguous, fragmented, and stylistically diverse nature of true freehand sketches drawn by non-experts.
This concern directly undermines the generalizability and practical applicability of the paper's core claim. If the method only performs well on "clean" contour maps but fails on authentic, casual sketches, its contribution of "reconstruction from hand-drawn sketches" is significantly weakened. The paper provides no evidence of the model's robustness to the challenges of real-world sketches.
（4）The framework competently integrates existing, well-established modules (ViT, PointNet, GCN). While the integration is sensible, it lacks a groundbreaking architectural or theoretical innovation. The primary contributions lean more towards solid engineering and a valuable dataset, rather than a significant advancement in scientific principle.

**Questions:**

Q1：What was the rationale behind selecting models designed for natural images (HMR, SPIN, etc.) as primary baselines, instead of focusing on a deeper, more exhaustive comparison against sketch-specific methods like SketchBodyNet? To what extent do you attribute the poor performance of these image-based methods to domain mismatch versus the absolute superiority of your method? Could you provide comparison results on more challenging, diverse, and authentic freehand sketches?
Q2：Could you elaborate in more detail on the specific drawing process for the "real hand-drawn sketches" in the Sketch3DS dataset? Were participants tracing clear contour images of 3D models, or drawing freely? Does the dataset include sketches with scribbled lines, disproportional body parts, or missing structural details? This is crucial for assessing the model's practicality in real-world scenarios.
Q3: What is the robustness of your model to sketch styles vastly different from those in the training set (e.g., minimalistic stick figures or child-like drawings)? Are there any experiments or plans to validate this aspect?
Q4:: What was the rationale behind selecting models designed for natural images (HMR, SPIN, etc.) as primary baselines, instead of focusing on a deeper, more exhaustive comparison against sketch-specific methods like SketchBodyNet? To what extent do you attribute the poor performance of these image-based methods to domain mismatch versus the absolute superiority of your method? Could you provide comparison results on more challenging, diverse, and authentic freehand sketches?

---

### Meta-Review · Area_Chair_Vgc3 · 2026-01-01

**Summary:**

The authors did not provide a rebuttal or engage in the discussion after the reviews were released. The scores (4442) indicate a consistent recommendation for rejection.

The AC identifies key limitations, including insufficient baseline comparisons with sketch-specific models and concerns about the quality and diversity of the Sketch3DS dataset, which undermine practical applicability. Additionally, the framework lacks significant architectural or theoretical innovations, focusing more on engineering than scientific advancement.

Overall, this paper falls short of ICLR acceptance standards, and the authors are encouraged to incorporate all feedback into future versions.

**Reviewer Concerns:**

No rebuttal was provided, so no concerns were addressed.

**Reviewer Scores:**

No rebuttal and no discussion likely mean that all the reviewers will maintain their ratings.

---

### Decision · Program_Chairs · 2026-01-26

Reject